**Data Availability Statement:** Anonymized data are available from Dr. Mingzhu Su (mzsu@aging.org. cn) upon reasonable request because we don't have the participant's consent to publicly release

# Patient-provider relationships in China: A qualitative study on the perspectives of healthcare students and junior professionals

Yuxian Du[1,2,3], Yan Du[4], Nengliang Yao🔘[3,5]*

**1** Hutchinson Institute for Cancer Outcome Research (HICOR), Fred Hutchinson Cancer Research Center, Seattle, Washington, United States of America, **2** Data Generation and Observational Studies, Bayer Healthcare U.S. LLC, Whippany, New Jersey, United States of America, **3** School of Health Care Management, Shandong University, Jinan, Shandong, People's Republic of China, **4** School of Nursing, the University of Texas Health Science Center at San Antonio, San Antonio, Texas, United States of America, **5** School of Medicine, University of Virginia, Charlottesville, Virginia, United States of America

* ayao@sdu.edu.cn

## Abstract

### Background

Mistrust and conflicts in patient-provider relationships (PPR) have become prevalent in China. The frequency of verbal and physical violence against healthcare workers has been increasing, but few interventions seem to be effective. Limited prior research has focused on the perspectives of healthcare professionals in training. This paper aimed to understand their viewpoints and conceptualize potentially actionable areas for future policy interventions.

### Methods

We analyzed de-identified training registration data of a convenience sample of 151 healthcare students and 38 junior professionals from 20 provinces in China. One open-ended question in the registration form asked the participant to comment on PPRs in China. We used qualitative thematic coding to analyze the narrative data. All answers were categorized into three overarching frames: patients, providers, and external agencies/regulations. Frequently mentioned themes in each frame were evaluated to generate an overall theoretical framework.

### Findings

Although fewer than 25% indicated that current PPRs are "good" or acceptable, 98% of respondents were optimistic about the future improvement of these relationships. The leading factors of PPRs mentioned as patient-relevant were eroding trust in the physician, unrealistic expectations, and ineffective communication. The provider-relevant themes highlighted were poor service quality, ineffective communication, and heavy workload. Leading themes relevant to external agencies or regulations were dysfunctional administration system, negative media reports, and disparity in healthcare resource distribution.

the original data. Data are stored in the secured cloud and local data warehouse.

**Funding:** This study was supported by the Fred Hutchinson Cancer Research Center in the form of a stipend for YD. YD also received consulting fees from the University of Washington School of Dentistry. The specific roles of these authors are articulated in the 'author contributions' section.

**Competing interests:** The authors have read the journal's policies and the authors of this manuscript have the following competing interests: "YD works as a contractor for Bayer Healthcare U. S. LLC but does not receive any income directly from Bayer. There are no patents, products in development or marketed products to declare. This does not alter our adherence to PLOS ONE policies on sharing data and materials.

**Abbreviations:** CNY, Chinese Yuan; PPR, Patient-Provider Relationship.

## Interpretation

Healthcare professionals in training had a negative view of the current situation but had confidence in future improvement. Patient, provider, and societal factors all contributed to the tension between patients and providers. All aspects of the healthcare sector should be carefully considered when contemplating policy or social interventions to improve the patient-provider relationship.

## Introduction

The World Health Organization reported that incivility and violence against health workers are prevalent all over the world [1]. Up to 38% of healthcare staff experienced physical violence at work [1]. Phillips found that the problem has been tolerated and generally ignored [2]. Violence against healthcare providers in China is increasing [3], garnering substantial media attention globally [4–6]. It was reported that healthcare providers experienced about 17,000 incidents of physical violence in 2010 [7]. Recent national surveys in China reported that 42.2 to 83.3% of healthcare workers experienced workplace violence [8–10].Though inconsistencies in the existing data are possible, the *Lancet* described the problem of incivility and violence against healthcare providers as a "crisis" for healthcare practices in China [11].

China's healthcare system provides services to a fifth of the world's population. Improvements in healthcare contributed partly to the increase in life expectancy from 43 to 76 years between 1960 and 2018 [12]. However, patient-provider relationships in China appeared to deteriorate after economic and healthcare reforms in the 1980s [3, 13, 14]. The current healthcare system faces challenges in delivering affordable and quality care with an acceptable bed-side manner [15, 16]. Ironically, the public-hospital-centered health system is mostly profit-driven [15]. Healthcare providers are incentivized to prescribe unnecessary tests and treatments [17]. The fee-for-service model misaligned the interests of hospitals and patients. Clinicians are believed to be pursuing self-interest over patient wellbeing. Chinese patients seeking care at tertiary level hospitals sometimes have unrealistic expectations for tertiary care [3].

The government and hospitals have taken measures, such as tighter security and zero mark-up drug policies, to improve patient-provider relationships and stop incivility and violence against healthcare providers [18]. However, the underlying systemic problems have not been adequately addressed. Incivility and violence against providers were found to be related to providers' lack of attention for patients because the providers feel disappointed with their healthcare occupation, turnover intention, and intention to leave their profession [19]. Researchers were concerned that deteriorating patient-provider relationships would affect the supply of the healthcare workforce [20, 21]. However, little information is found in the literature that describes the patient-provider relationship in China from the perspective of healthcare professionals in training.

The purpose of this investigation was to qualitatively examine the viewpoints of healthcare students and junior professionals on patient-provider relationships in China. It gives a voice to providers in training to express their feelings and concerns and ensures that the study findings of providers' relationships with patients are grounded in their experiences. We also discuss potentially actionable policy interventions in this study.

## Materials and methods

Shandong University Cheeloo College of Medicine organized the first training of home-based health care in China in early August of 2018. The call for registration was sent through China's

most popular messaging app WeChat to healthcare personnel in June and July of 2018. The training was free to healthcare students, while junior healthcare professionals paid 2000 Chinese Yuan (CNY) each for the registration. A total of 189 healthcare students and junior professionals from 20 provinces submitted registration forms between June 29 and July 25 of 2018. The registration form included one open-ended question asking about participants' views on the contemporary patient-provider relationship in China. The form also had questions about their basic demographics and their employer or institution of study. Since de-identified registration data was used in this study, the ethics committee of Shandong University waived the need for ethical approval and the requirement for consent.

We used thematic analysis to summarize the responses; two qualified researchers independently conducted the thematic coding. Both researchers 1 and 2 received doctoral and post-doctoral training in qualitative and mixed-methods research and have published peer-reviewed works on qualitative research. Researcher 1 reviewed responses one participant at a time and generated codes and themes for each answer until saturation (i.e., no new codes or themes emerged). In our study, saturation was reached by analyzing responses of 4 junior professionals and 16 healthcare students. Researcher 2 reviewed all responses to the question on the patient-provider relationship, generated codes and themes for each response, categorized all answers into multiple thematic groups, and calculated the frequency of appearance for each theme. Both researchers decided that opinions should be categorized into those describing the current situation of patient-provider relationships (PPR) and those providing possible reasons for the current situation.

Regarding descriptions of current PPRs, researcher 1 defined the current status and trends, while researcher 2 defined three aspects, including the current status, trends, and hopefulness that the state of patient-provider relationship will improve. For example, phrases such as "*buhao*" (not good) and "*jingzhang*" (intense) were coded as negative. As for possible reasons for the current status of PPRs, research 1 identified four overarching thematic groups containing 30 itemized codes, while researcher 2 identified three overarching thematic groups containing 43 itemized codes. Both researchers then discussed the coding structure and derived a reconciled summary, which assessed the factors impacting PPRs from three perspectives: patient, provider, and societal.

We assessed group characteristics differences using Fisher's exact tests and summarized the relative weights of opinion using descriptive statistics. For geographical regions, we used Chinese geographical regions: North (provinces "Beijing", "Tianjin", "Hebei", "Shanxi", "Inner Mongolia"), Northeast (provinces "Liaoning", "Jilin", "Heilongjiang"), Northwest (provinces "Shaanxi", "Gansu", "Qinghai", "Ningxia", "Xinjiang"), East (provinces "Shanghai", "Jiangsu", "Zhejiang", "Anhui", "Fujian", "Jiangxi", "Shandong"), South Central (provinces "Henan", "Hubei", "Hunan", "Guangdong", "Guangxi", "Hainan", "Hong Kong", "Macau"), and Southwest (provinces "Chongqing", "Sichuan", "Guizhou", "Yunnan", "Tibet"). Because only small numbers of participants came from Northeast, Northwest, and Southwest regions, we regrouped some regions together: (1) North, Northeast, and Northwest as "North"; (2) South Central and Southwest as "South." We conducted all quantitative analyses in SAS 9.3 (Cary, NC), and performed thematic coding in Microsoft Excel (Redmond, WA).

## Results

### Participant characteristics

189 healthcare students and junior professionals answered the question about patient-provider relationships, and Table 1 summarizes their characteristics. About four-fifths of the respondents were healthcare students (n = 151), with more than ninety percent from academic

**Table 1. Geographical and professional information on survey respondents.**

| Variables | Categories | Junior Professionals (n = 38) | | Healthcare Students (n = 151) | | Fischer's Exact Test p-value |
|---|---|---|---|---|---|---|
| | | Count | % | Count | % | |
| **Profession** | Hospital Management | 12 | 31.6% | 47 | 31.1% | 0.972 |
| | Medicine/Rehabilitation | 9 | 23.7% | 33 | 21.9% | |
| | Nursing | 17 | 44.7% | 71 | 47.0% | |
| **Sector** | Universities | 8 | 21.1% | 140 | 92.7% | <0.001 |
| | Hospitals | 13 | 34.2% | 9 | 6.0% | |
| | Others/Unknown | 17 | 44.7% | 2 | 1.3% | |
| **Geographical Region** | East | 13 | 34.2% | 95 | 62.9% | 0.001 |
| | North | 7 | 18.4% | 38 | 25.2% | |
| | South | 13 | 34.2% | 18 | 11.9% | |

institutions. About 47% of the healthcare students and 45% of junior professionals were in nursing. More than 60% of the medical students were from the eastern part of China, where there are disproportionately more medical colleges than in any other geographical region. Over two-thirds of junior healthcare professionals were working in the eastern or southern part of China, which is more economically developed than other areas (some junior professionals chose not to disclose their geographical regions).

## Views on the current patient-provider relationships (PPR)

Two researchers examined the questionnaire results describing participants' views on the current PPR (Table 2). According to results from Researcher 2, about 12% of the survey responses indicated that the current PPR is good or ideal, while 48% reported that they feel the state of PPRs is trending in a positive direction. Almost all survey participants who mentioned hopefulness about PPRs reported positively.

## Views on reasons for the current situation

In addition to providing a general description of the current situation of PPR, the survey respondents also shared opinions on possible causes of current tensions between patients and providers. As the PPR is an evolving and dynamic social contractual relationship, it receives impacts from multiple aspects. Here we summarize the results, categorized by the patient, provider, and societal perspectives and influences. These perspectives were not directly from the patient, provider, and society, but opinions expressed by the survey participants (i.e., healthcare students or junior professionals), speaking from their knowledge or experience.

**Table 2. Quantitative summary of views on current patient-provider relationship (PPR).**

| | | Researcher 1 | | Researcher 2 | |
|---|---|---|---|---|---|
| | | Ref count | Ref % | Ref count | Ref % |
| **PPR (% positive)** | Current Status | 2/9 | 22% | 16/133 | 12% |
| | Trend | 2/4 | 50% | 26/54 | 48% |
| | Hopefulness | N/A | N/A | 56/57 | 98% |

Note: Researcher 1 used a "coding until saturation" method, which did not use all the survey responses. That is why the overall counts of codes are different between Researcher 1 and 2.

**Table 3. Reasons for current PPR–Patient's influence.**

| | Researcher 1 | | Researcher 2 | |
|---|---|---|---|---|
| | **Ref count [a]** | **Ref %** | **Ref count [b]** | **Ref %** |
| **Patient's Influence** | | | | |
| Ineffective Communication | **3** | **27%** | 26 | 12% |
| Lack of Empathy | 2 | 18% | 22 | 10% |
| Unrealistic Expectation | 2 | 18% | 48 | 22% |
| Information Acquisition | 1 | 9% | N/A | N/A |
| Morality | 1 | 9% | N/A | N/A |
| Lack of Trust in Physicians | 1 | 9% | **56** | **25%** |
| Complexity of Human Nature | 1 | 9% | N/A | N/A |
| Lack of Respect | N/A | N/A | 11 | 5% |
| High Cost Burden | N/A | N/A | 21 | 10% |
| Lack of Access | N/A | N/A | 36 | 16% |
| **Subtotal** | 11 | 100% | 220 | 100% |

Notes:

a. Ref = Reference, each count meaning the one participant mentioned this in his/her response.

b. For coding from Researcher 2, if a single participant mentioned one theme multiple times in his/her response, we would only count it as once since they all came from one participant.

From the perspective of the responded healthcare students and junior professionals, the patient's influence was mainly from "ineffective communication" and "lack of trust in physicians," as identified in Table 3. "Ineffective communication" (i.e., patients not willing to communicate with the physicians) represents 27% of the patient-related opinion as coded by researcher 1, and "lack of trust in physicians" represents 25% of the patient-related opinion as coded by researcher 2. Ineffective communication, as defined by the survey respondents, was either not having enough time to communicate with the provider during physician visits or lacking necessary medical information. Hence, patients found it challenging to comprehend all information given by their healthcare provider. Lack of trust referred to (1) patients' fear of excessive prescription from the provider or (2) selectively trusting clinicians with richer experiences. Some representative quotes were:

- Communication:

  ○ *"Patients and physicians are not standing in each other's shoes."*

  ○ *"Communication was neither timely nor effective."*

  ○ *"Patients and their relatives are not communicating with the doctors."*

- Trust:

  ○ *"They have to use their social network or give a red envelope to receive the medical services they are entitled to."*

  ○ *"The patients would use the information they see from the internet to defend against the physicians."*

Other leading patient-related factors contributing to the current status of PPRs included lack of empathy, unrealistic expectations, information acquisition, morality, the complexity of human nature, lack of respect, high cost burden, and lack of access. "Lack of empathy" refers to

the view that some patients were believed not to understand the hardships or challenges physicians face in their profession. Some patients had unrealistic expectations of medical services, hoping physicians could address all illnesses with one single visit. As the internet became increasingly accessible, patients acquired medical information online and used it against clinicians' diagnoses and recommendations. Patients' morality and lack of trust were also a concern of the survey respondents, because "some patients have relatively lower moral standards and lack basic respect for the clinicians." Some responses mentioned the complexity of human nature, meaning that patients had additional expectations for their physicians aside from merely curing the disease, creating greater complexity in the patient-provider relationship. Other survey respondents also mentioned the cost burden on patients, as well as lack of healthcare access, especially for patients in rural or underdeveloped areas.

The most prevalent influences from the provider related to service quality and ineffective communication, as shown in Table 4. According to the survey respondents, service quality was limited by the attitudes of clinicians and the encounter time. Researcher 1 found that 31% of provider-related opinions were about service quality, while researcher 2 found 24%. The "ineffective communication" component encompassed both the physician's capability of communicating complex medical information to patients and restrictions related to the short amount of encounter time per patient. Researchers 1 and 2 found that this component comprised 23% and 16% of the provider-related opinion, respectively. Representative quotes included:

- *"Some places have bad service quality."*

- *"Physicians have limited time per patient, so they do not have enough time to talk."*

- *"Large hospitals have a strong siphon effect that attracts many physicians."*

- *"Rural areas lack healthcare resources."*

Besides, other influential factors included lack of empathy, heavy workload, lack of trust, the complexity of human nature, an insufficient workforce, morality, information asymmetry,

**Table 4. Reasons for current PPR–Provider's influence.**

| | Researcher 1 | | Researcher 2 | |
|---|---|---|---|---|
| | Ref count [a] | Ref % | Ref count [b] | Ref % |
| **Provider's Influence** | | | | |
| Ineffective Communication | **3** | **23%** | **46** | **16%** |
| Lack of Empathy | 2 | 15% | 21 | 7% |
| Poor Service Quality (Bad Attitude / Limited Encounter Time) | **4** | **31%** | **69** | **24%** |
| Heavy Workload | 2 | 15% | 30 | 11% |
| Lack of Trust | 1 | 8% | 21 | 7% |
| The complexity of Human Nature | 1 | 8% | N/A | N/A |
| Insufficient Workforce | N/A | N/A | 30 | 11% |
| Morality | N/A | N/A | 20 | 7% |
| Information Asymmetry | N/A | N/A | 19 | 7% |
| Psychological Pressure | N/A | N/A | 15 | 5% |
| Monetary Incentives | N/A | N/A | 12 | 4% |
| **Subtotal** | 13 | 100% | 283 | 100% |

Notes:

a. Ref = Reference, each count meaning the one participant mentioned this in his/her response.

b. For coding from Researcher 2, if a single participant mentioned one theme multiple times in his/her response, we would only count it as once since they all came from one participant.

**Table 5. Reasons for current PPR–Societal influence.**

| | Researcher 1 | | Researcher 2 | |
|---|---|---|---|---|
| | Ref count [a] | Ref % | Ref count [b] | Ref % |
| **Societal Influence** | | | | |
| Insufficient Healthcare Resource | 1 | 14% | N/A | N/A |
| Dysfunctional Administration System | **2** | **29%** | **35** | **33%** |
| Morality (Yi Nao, etc.) | 1 | 14% | 15 | 14% |
| Negative Media Reports | 1 | 14% | 23 | 21% |
| Disparity in Healthcare Resource Distribution | 1 | 14% | 21 | 20% |
| The complexity of Human Nature | 1 | 14% | N/A | N/A |
| Insufficient Payment Share (i.e., Insurance) | N/A | N/A | 13 | 12% |
| **Subtotal** | 7 | 100% | 107 | 100% |

Notes:

a. Ref = Reference, each count meaning the one participant mentioned this in his/her response.

b. For coding from Researcher 2, if a single participant mentioned one theme multiple times in his/her response, we would only count it as once since they all came from one participant.

psychological pressure, and monetary incentives. Survey respondents mentioned that an insufficient workforce in the Chinese healthcare sector could be a reason for the heavy workload of physicians and their psychological pressure. There seemed to be more demand for healthcare from patients than what the healthcare system could supply. Clinician factors that may contribute to tensions in patient-provider relationships were morality, lack of empathy, and trust. According to the survey respondents' view, some physicians were believed to have relatively lower moral standards and cared more about their financial gain than patient wellbeing. Others mentioned that physicians' income was inadequate concerning their training and expertise, so monetary incentives such as red envelopes and pharmacy rebates could have driven physicians' behavior.

In terms of societal influences (Table 5), most opinions pointed toward a "dysfunctional administration system." Researcher 1 and 2 found that societal-related opinions mentioned this component at 29% and 33%, respectively. According to the responses, establishing an improved system with more effective regulations would be critical to mitigating the current tension between patients and providers. Representative quotes included:

- *"Under the current system, public hospitals are profit-driven."*

- *"Healthcare system is a dynamic structure, and we need to change many aspects."*

Participants mentioned other societal influences. Deficiencies in healthcare resources and distribution disparities were two of the major topics in the responses. The negativity of media reports and their emphasis on medical disruption (*Yi Nao*) inevitably encouraged more attacks on providers during any medical dispute. Others also attributed the tension between patients and providers to insufficient payment shares from insurance and social support, which cast more onerous financial burdens on both the patients and the healthcare providers.

## Discussions

A small percentage of healthcare students and junior professionals felt that the current patient-provider relationship was "good" or acceptable. However, many of the respondents were optimistic about the future improvement of the relationship. The leading explanations of the

current PPR mentioned as patient-relevant were eroding trust in the physician, over-expectations, excessive demand for healthcare, and miscommunication. The provider-relevant themes highlighted were miscommunication and lack of a proper bedside manner. Leading themes relevant to external agencies or regulations were biased reports from social media, flaws with the healthcare delivery and payment system, disparities in healthcare resource distribution, and negative social impacts from medical disputes.

One response effectively summarized the opinions of healthcare students and junior professionals on the current patient-provider relationship—"miserable but hopeful." They had a negative view of the current situation but had confidence in future improvement. Factors related to the patient, provider, and external agencies all contributed to the tension between patients and providers. Moreover, many of these factors interact with each other. For example, the mostly fee-for-service payment system with low pay rates drives patients without complicated conditions to seek care in higher-tiered hospitals, and providers, in turn, to deliver care lacking adequate bedside manner to an excessive number of patients. Thus, no single silver bullet can completely solve the patient-provider dispute. All aspects of the healthcare sector should be carefully considered when contemplating policy or social interventions.

Trust needs to be rebuilt in the Chinese healthcare system [22]. As Confucius proclaimed 2600 years ago, "people cannot stand without trust." Similarly, patient-provider trust is the foundation of healthcare [14, 22]. On the patient side, some emotionally-suffering patients viewed them as the disadvantaged and thus had less trust in their providers [20]. While some socioeconomically-advantaged individuals viewed themselves as entitled and thus being arrogant and not trusting their physicians [23]. In some scenarios, their lack of trust in providers could be addressed by improving the patient-perceived quality of care [24]. On the provider side, healthcare professionals and hospitals have become profit-driven after the health reform [15], and industry payments and monetary gifts from patients have accounted for a considerable portion of Chinese providers' salaries [25]. The performance evaluation systems in hospitals need to be reformed to focus more on quality of care rather than commercial revenues. Though the Commission on Health and Family Planning in China wants to realign incentives [26], China has a mostly fee-for-service single-payer system with strict price control policies [16]. The development of private healthcare and commercial insurances may help supplement the income of healthcare providers [27, 28]. Non-government non-profit healthcare could play a role in changing the pattern and revenue model of quality healthcare services [29].

Pre- and post-acute care are underdeveloped in China; development may help redistribute the workloads of healthcare professionals from different levels of facilities [16]. The three-tier healthcare system does not meet the increasing demands of a rapidly aging society. Rather than merely being perceived as functioning differently compared to tertiary care, first- and second-tier healthcare facilities are considered of lower quality and less value. Furthermore, patients often have unrealistic expectations of third-tier or tertiary care. Patients do not need a referral from first- or second-tier healthcare facilities before visiting a third-tier one, causing third-tier facilities to attract a disproportionate share of patients and resources over time. However, secondary care facilities also provide acute care. We suggest that the Chinese government considers modifying the three-tier grading system and building a healthcare system based on the episodes of care or the continuum of healthcare. Instead of grading the healthcare facilities, the health planners can assign facilities to different roles across the continuum of care. The current first- and second-tier healthcare institutions can be incentivized to provide more pre- and post-acute care, rather than competing with large comprehensive medical centers and aiming for a higher-level grading. By spreading out the workload, providers at tertiary care facilities can spend more time communicating with patients and family, reducing information asymmetry, and improving the patient-provider relationship.

It is believed that the Chinese government missed an opportunity to educate the public about health and healthcare via public media and the education system since the 1980s [30, 31]. False or misleading health information "exploded" in state-controlled media and the internet since the 2000s. Chinese residents have low literacy in health and healthcare, which is highly correlated with insufficient education, rural residence, low-pay occupations, and lower overall household income. These characteristics happen to coincide with the socio-demographics of the violent offenders in healthcare [19, 21, 32]. Additionally, academic medical systems have not done their fair share to educate the public about health and health care services [33]. We argue that government healthcare agencies have responsibilities to disseminate truthful information to patients and communities, which may reduce excessive demands and over-expectations of tertiary care.

This study has some limitations. First, we used a convenience sample without knowing how many have seen the initial recruitment notice, and everyone who answered the open-ended question was interested in home health for the elderly population, so their opinions may be biased and unrepresentative of a broader proportion of the population. Second, our analyses were only based on one single open-ended question, but the participants' responses encompassed multiple domains, and their weights were unbalanced. Third, though 189 individuals might be a sufficient sample size for qualitative analysis, the amount of information provided by each respondent varied, so the overall data quantity may be less than needed for saturation. Fourth, our analytical approach was different from a traditional qualitative approach; we aimed to provide insights into patient-provider relationship issues and present the relative frequencies of reasons for these issues reported by healthcare students and junior professionals. Future qualitative research is needed to establish a more comprehensive theoretical framework on the patient-provider relationship, and possible approaches include triangulation, crystallization, and grounded theory. Fifth, this was a cross-sectional analysis without retrospective and prospective information, so longitudinal trends were not available. Sixth, this study only investigated the perspectives from healthcare students and junior professionals, which could not represent the perspectives from patients, senior healthcare providers, health administrators and regulators, or any other stakeholders in the healthcare sector. Future studies should seek a more comprehensive understanding of the patient-provider relationship from various perspectives.

## Conclusions

Healthcare students and junior professionals had a negative view of the current patient-provider relationship but had confidence in future improvement. Patient, provider, and societal factors all contributed to tensions between patients and providers. Non-government non-profit healthcare could contribute to realigning incentives for healthcare providers. Hospital regulation systems should consider offering incentives for the current first- and second-tier healthcare systems to provide more pre- and post-acute care. Thus, the workload of tertiary care workers can be reallocated, which may help improve patient-provider communication and bedside manner. The government and academic medical systems need to step up in educating the public about health and health care services. Additional research is needed to test these practical and policy propositions.

## Acknowledgments

We sincerely thank all the participants for sharing their views on the patient-provider relationships in China.

## Author Contributions

**Conceptualization:** Nengliang Yao.

**Data curation:** Yuxian Du, Yan Du, Nengliang Yao.

**Formal analysis:** Yuxian Du, Yan Du.

**Investigation:** Yuxian Du, Yan Du, Nengliang Yao.

**Methodology:** Yuxian Du, Yan Du, Nengliang Yao.

**Project administration:** Yuxian Du, Nengliang Yao.

**Supervision:** Nengliang Yao.

**Validation:** Yuxian Du, Yan Du.

**Visualization:** Yuxian Du.

**Writing – original draft:** Yuxian Du, Nengliang Yao.

**Writing – review & editing:** Yuxian Du, Yan Du, Nengliang Yao.

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
