## [Decision Letter · Decision Letter 0]

9 Jul 2020

PONE-D-20-17587

Patient-Provider Relationships in China: A Qualitative Study on the Perspectives of Healthcare Students and Junior Professionals

PLOS ONE

Dear Dr. Yao,

Thank you for submitting your manuscript to PLOS ONE. After careful consideration, we feel that it has merit but does not fully meet PLOS ONE’s publication criteria as it currently stands. Therefore, we invite you to submit a revised version of the manuscript that addresses the points raised during the review process.

ACADEMIC EDITOR: Please insert comments here and delete this placeholder text when finished. Be sure to:

Provide specific responses to the reviewers and editor, particularly reviewer 2's. 

We look forward to receiving your revised manuscript.

Kind regards,

Lanjing Zhang, MD, MS

Academic Editor

PLOS ONE

Journal Requirements:

We note that one or more of the authors are employed by a commercial company: "Bayer Healthcare US LLC,"

4. We note you have included a table to which you do not refer in the text of your manuscript. Please ensure that you refer to Tables 1 and 2 in your text; if accepted, production will need this reference to link the reader to the Table.

Additional Editor Comments (if provided):

This interesting study addresses a critical problem in China, but there are some methodological concerns. Please be mindful that unsatisfactory responses to the comments from reviewers and the editor may result in rejection to the revision. In other words, the chance of accepting the revised manuscript is very small unless very significant changes are made.

Major points:

1. The sample size is small, which mandate smaller/fewer subgrouping. Thus, Table 1 may need to re-group some of the categories.

2. Fisher exact test should be used in Table 1 since some cells have a number smaller than 5.

3. Please provide how many people received or read the advertisement so that we will better understand the potential selection bias.

4. Were the training free? Were the participants compensated or provided other incentives?

5. In my understanding, there were only 13 participants who answered the questionnaires as shown by researcher 1. The sample number is very small. Further, if it is true, only about 8.5% (13/151) participants responded. The proportion of missing data appeared too big.

Minor points:

1. In tables, please provide a note indicating ref=reference.

2. Please indicate how many participants answered the questionnaire about patient physician relationship.

Reviewers' comments:

Reviewer's Responses to Questions

**Comments to the Author**

1. Is the manuscript technically sound, and do the data support the conclusions?

Reviewer #1: Yes

Reviewer #2: Partly

2. Has the statistical analysis been performed appropriately and rigorously? 

Reviewer #1: Yes

Reviewer #2: N/A

3. Have the authors made all data underlying the findings in their manuscript fully available?

Reviewer #1: Yes

Reviewer #2: Yes

4. Is the manuscript presented in an intelligible fashion and written in standard English?

Reviewer #1: Yes

Reviewer #2: Yes

5. Review Comments to the Author

Reviewer #1: This study analyze the Patient-Provider Relationships in China from the perspectives of

healthcare students and junior professionals. It will be better if following issues could be fixed.

1.There are 151 healthcare students and 38 junior professionals from 20 provinces in China. What is the representativeness of sample? Is there difference of Patient-Provider relationships between healthcare students and junior professionals?

2.Geographical Region in Table 1 is “east,south and north”. What is the basis of regional division?

In general, the region is divided into “east,central and west” or “south,central and north”.

Reviewer #2: 1. Some statements in Introduction need to be supported by more recent data, such as in line 6, the first paragraph in Introduction, literature 7.

2. The second paragraph in Introduction need to be more detailed about why patients-provider relationship is relatively serious.

3. Some words in this article is not spelling correctly.

4. In my opinion, health care provider should be workers in hospital or health institution. However, respondents in this study are most students. The views of healthcare professional on PPR are different from those of students, personally thinking. Why did the author think the opinion of students could represent patient-provider relationship?

5. The professional quality of two researchers need to be illustrated in Method.

6. The Patient’s influence is stilled answered by Providers. Answer from patients may be more reliable, considering the focus of this article is patient-provider relationship.

6. PLOS authors have the option to publish the peer review history of their article (what does this mean?). If published, this will include your full peer review and any attached files.

Reviewer #1: No

Reviewer #2: No

---

## [Author Response · Author response to Decision Letter 0]

25 Aug 2020

We appreciate the opportunity to address the comments and revise our manuscript. Please find item-by-item responses to the comments in our response letter. Thanks!

---

## [Editor Report · Decision Letter 1]

2 Oct 2020

Patient-Provider Relationships in China: A Qualitative Study on the Perspectives of Healthcare Students and Junior Professionals

PONE-D-20-17587R1

Dear Dr. Yao,

We’re pleased to inform you that your manuscript has been judged scientifically suitable for publication and will be formally accepted for publication once it meets all outstanding technical requirements.

Kind regards,

Lanjing Zhang, MD, MS

Academic Editor

PLOS ONE

Additional Editor Comments (optional):

Thank you for your careful revision!
---

## [Editor Report · Acceptance letter]

12 Oct 2020

PONE-D-20-17587R1 

Patient-provider relationships in China: A qualitative study on the perspectives of healthcare students and junior professionals 

Dear Dr. Yao:

I'm pleased to inform you that your manuscript has been deemed suitable for publication in PLOS ONE. Congratulations! Your manuscript is now with our production department. 

Kind regards, 

on behalf of

Dr Lanjing Zhang 

Academic Editor

PLOS ONE